# Fatty Acid and Stable Carbon Isotope Composition of Slovenian Milk: Year, Season, and Regional Variability

**DOI:** 10.3390/molecules25122892

**Published:** 2020-06-23

**Authors:** Doris Potočnik, Lidija Strojnik, Tome Eftimov, Alenka Levart, Nives Ogrinc

**Affiliations:** 1Department of Environmental Sciences, Jožef Stefan Institute, Jamova cesta 39, 1000 Ljubljana, Slovenia; doris.potocnik@ijs.si (D.P.); lidija.strojnik@ijs.si (L.S.); 2Jožef Stefan International Postgraduate School, Jamova cesta 39, 1000 Ljubljana, Slovenia; tome.eftimov@ijs.si; 3Computer Systems, Jožef Stefan Institute, Jamova cesta 39, 1000 Ljubljana, Slovenia; 4Department of Animal Science, Chair of Nutrition, Biotechnical Faculty, Groblje 3, 1230 Domžale, Slovenia; alenka.levart@bf.uni-lj.si

**Keywords:** milk, fatty acids, stable carbon isotope composition, GC-FID, CSIA, statistical analyses

## Abstract

This study examined the percentage and stable isotope ratios of fatty acids in milk to study seasonal, year, and regional variability. A total of 231 raw cow milk samples were analyzed. Samples were taken twice per year in 2012, 2013, and 2014, in winter and summer, covering four distinct geographical regions in Slovenia: Mediterranean, Alpine, Dinaric, and Pannonian. A discriminant analysis model based on fatty acid composition was effective in discriminating milk according to the year/season of production (86.9%), while geographical origin discrimination was less successful (64.1%). The stable isotope composition of fatty acids also proved to be a better biomarker of metabolic transformation processes in ruminants than discriminating against the origin of milk. Further, it was observed that milk from Alpine and Mediterranean regions was healthier due to its higher percentage of ω-3 polyunsaturated fatty acid and conjugated linoleic acid.

## 1. Introduction

There is an increasing interest in the quality of foodstuffs, and agricultural research continues to focus on improving the nutrient profile of food products. Milk is a valuable commodity that is consumed directly, as well as being a minor and major ingredient in many food products [1]. It is also rich in nutrients, a resource of proteins (2.6–4.2%), vitamins (0.1%), minerals (0.7%), and fatty acids (95–96% of total milk fat), and is considered by many as an essential part of a healthy diet [2,3]. In Europe, milk provides 9% of the dietary energy supply, 19% of the dietary protein supply, and between 12–14% of dietary fat supply [4].

According to the Agriculture and Rural Development, Analysis and Outlook Unit European’s farms produced 172.2 million tonnes of raw milk in 2018, a number that they expect to increase to 179 million tonnes by 2030, but at a slower pace (+0.6%) than in 2008–2019, and represents a year-on-year increase of 1.6 million tonnes (+1% per year on average) [5]. In Slovenia, dairy farmers produced 564,000 tons of cows’ milk in 2019 (1.2% less than in 2018), and 164,000 tons of drinking milk, which represents a decrease of more than 2% compared to 2018. Despite the decrease, imports increased by almost 13% [6]. Consumers prefer milk with favorable sensory qualities, such as flavor, which depends on animal management factors, including breed, lactation stage, processing techniques, and diet regime. However, diet is considered the primary and the most sensitive factor responsible for creating a specific milk flavor [7] and also a significant factor affecting its fatty acid (FA) composition. These differences also reflect differences in regional dairy production systems, and linking a milk product to a specific region is of high importance as a recognition of quality [1]. Quality linked to geographical origin has created significant interest in building local and regional food systems across Europe, including Slovenia. 

Lipid content and FA profile are considered to have an important role in the human diet [7,8]. For example, certain milk FAs, such as linoleic and α-linolenic FAs (C18:2ω-6 and C18:3ω-3), are essential and cannot be synthesized by the human body [2]. Further, a lower proportion of saturated fatty acids (SFA) and a higher proportion of unsaturated fatty acids (UFA), especially polyunsaturated fatty acids (PUFA), is desirable [9]. The consumption of SFA (especially lauric (C12:0), myristic (C14:0), and palmitic acids (C16:0)) [9] is associated with cardiovascular disease, while UFAs are considered as beneficial for human health. Other functional food components are omega-3 polyunsaturated FA (ω-3 PUFAs), which improve neurological functions and are essential in early human development years, and conjugated linoleic acid (CLA), which has anti-carcinogenic, anti-diabetic, anti-atherogenic effects [10,11]. 

Farmers can significantly alter the FA composition of milk by managing the nutrition of their dairy cows [12]. Fresh forages (especially grass) in the diet enhance the concentration of unsaturated FA, such as α-linoleic acid and CLA, and decrease the levels of SFA, while adding grass conservation products (hay or silage) increases the proportion of SFA in milk [13]. Besides, the total SFA concentration of milk fat is usually lower in summer compared to winter due to the higher consumption of summer forage crops that have a higher content of PUFAs. The higher content of monounsaturated fatty acids (MUFA) and PUFA in milk from cows grazing in highland regions can be attributed to differences in the botanical composition since higher botanical diversity and energy shortage in cows grazing in mountain pastures modifies the bacterial population and lipid accumulation of UFAs in the rumen [14]. Fatty acid composition of milk fat has also been used as a criterion to detect adulteration of milk, such as the substitution of part of the fat or proteins of plant or animal origin and admixtures of milk of different species. The method to characterize the matrixes according to the animal species can be greatly improved when combined with the content of triacylglycerols [15]. 

The effects of feeding regime can also be tracked using stable isotope analysis using isotope ratio mass spectrometry (IRMS). In dairy products, the isotope composition is also primarily determined by the cows’ diet, as well as their metabolisms [16]. Differences in stable C isotope abundance is mainly the result of the different ingested proportion of feeds derived from C_3_ and C_4_ plants. The metabolism of C_4_ plants, such as corn or sugar, uses a different biosynthetic pathway to fix atmospheric CO_2_ compared to C_3_ plants (grass and cereals), resulting in the accumulation of the heavier ^13^C isotope compared to C_3_ plants, which leads to higher *δ*^13^C values. These differences also show up in the isotopic composition of FAs [17,18]. Since the spectrum of plants used for feeding purposes depends mainly on local soil and climatic condition, stable isotope composition can also encompass information about the geographical origin of dairy products useful for authenticity control. 

Up to now, studies have focused on stable isotope ratio measurements of *δ*^13^C, *δ*^15^N, and *δ*^34^S values in casein, *δ*^2^H, *δ*^18^O in milk water [19,20,21,22] and Sr in dry milk samples [21]. One of the most recent significant advances has been the development of compound-specific stable isotope analysis (CSIA) that enables the identification of relevant biomarkers, such as fatty acids, in food authenticity studies. Some studies have even looked at the correlation between long-chain FAs (α-linolenic FA-C18:3ω-3, eicosapentaenoic FA-C20:5ω-3) and stable isotope measurements to distinguish between organic and conventionally produced milk [17,23]. In another study, the authors investigated the relationship between the carbon isotope compositions of CLA in milk and fatty acids in different dietary sources (C_3_ versus C_4_ plants) to understand the transformation of FA from diet to milk [17]. The results suggested that the *δ*^13^C-change in CLA did not originate from the microbial biohydrogenation in the rumen, but most probably from endogenous desaturation of *t*11–18:1.

Only a few studies, however, have focused on *δ*^13^C and *δ*^2^H values in milk fatty acids. Ehtesham and co-workers [24] demonstrated that *δ*^2^H values in precipitation were highly correlated with the *δ*^2^H values in bulk milk and specific fatty acids (butyric, myristic, palmitic, and oleic acid). Separation of milk powder origin to geographic sub-regions within New Zealand has been achieved, with clear discrimination between South Island and North Island and with significant clustering of regions. Further, a linear discriminant analysis (LDA) model based on *δ*^2^H and *δ*^13^C values of butyric, myristic, palmitic, and oleic fatty acids has provided the best separation of samples from the Northern and Southern parts of New Zealand [1]. Ehtesham et al. [25] also focused on the relationship between *δ*^2^H and the FAs in milk and the water component of milk (feed and farm water). The authors found that the *δ*^2^H values of FAs in feed and *δ*^2^H values of farm water were related to the *δ*^2^H values of milk FAs, milk solids, and milk water and could be used as a biogeochemical marker surrogate to environmental conditions. This approach can also be applied to determine the dietary regime and geographical origin of milk products where the water has been removed (milk powder) or is altered in the case of processed milk products (cheese and butter).

The main objective of this study was to evaluate both the need and options for origin tracing, focusing on Slovenian milk. In 2016, the Slovenian Ministry of Agriculture, Forestry, and Food began a three-year project to promote and raise awareness of the importance and characteristics of locally produced and processed foods. During this time, the protective mark “Selected Quality” national scheme and the “Selected Quality-Slovenia” was established. Nowadays, most milk and dairy products, produced and processed in Slovenia, use the “Selected Quality - Slovenia” protective mark. This mark indicates that the products have the following distinguishing properties: (i) the origin of raw milk—the milk and other dairy products are produced entirely in the same country, Slovenia; (ii) milk quality—excellent microbiological quality of milk; (iii) freshness of raw milk—it’s 15 h at most from the takeover of milk at the collection site to its acceptance into the processing plant. Therefore, to protect the obtain mark and prevent possible fraud, a robust screening method to determine the authenticity and regional traceability of milk is needed. In this study, two methods (fatty acid composition and isotope ratios of carbon within fatty acids) were applied. Additionally, the effect of the production year and season on the FA composition of milk was also investigated. Finally, the FA profiles pertaining to the milk’s essential and beneficial components were also studied, and a comparative study was made to examine the differences in the quality of milk from different geographic regions in Slovenia.

## 2. Results and Discussion

### 2.1. Milk Fatty Acids Composition 

#### 2.1.1. Influence of the Year and Season of Production and Impact of Feed

The amount of individual fatty acids, as well as SFA, MUFA, PUFA, ω-3 PUFA, ω-6 PUFA, and ω-6:ω-3, according to year, season, and feeding regime (corn/non-corn) are given in Table 1. As expected, SFAs were the most abundant FA groups in milk, followed by MUFA and PUFA. The most abundant FA was C16:0, ΣC18:1 *cis* and *trans* isomers, C18:0, and C14:0, which is in agreement with the published data [26,27]. Time (year/season) was revealed as an important factor influencing fatty acid composition, but not all fatty acids were equally affected. The effect of the year of production (2012, 2013, 2014) on FA composition was investigated using the thirty-five FAs detected in both summer and winter. The results revealed a significant variation in the amount of FA with year except for C12:0, C13:0, C14:0, C14:1, C15:0, C15:0, C17:0, C18:2ω-6, C18:3ω-3, and C22:1ω-9. The year 2012 produced the lowest amount of ∑SFA (67.76 ± 2.49% of total FA) and the highest amount of ∑MUFA (27.26 ± 2.10% of total FAs) and CLA (rumenic acid-C18:2*c*9,*t*11; 0.86 ± 0.28% of total FAs). Furthermore, 2012 also had the lowest percentage of C16:0 (32.60 ± 2.13% of total FAs) and hypercholesterolemia FAs (HFA; C12:0 + C14:0 + C16:0) (48.13 ± 2.97% of total FA) and the highest percentage of *cis* and *trans* isomers C18:1ω-9 (23.7 ± 2.17% of total FA). 

The percentage of PUFA, ω-6 PUFA, and ω-3 PUFA was higher in 2013 (4.05 ± 0.98%, 2.61 ± 0.58%, 0.84 ± 0.34% of total FAs, respectively) compared to 2012 (3.69 ± 1.14%, 2.31 ± 0.24%, 0.70 ± 0.22% of total FAs, respectively) and 2014 (3.55 ± 0.56%, 2.30 ± 0.33%, 0.71 ± 0.21% of total FAs, respectively), while the ratio ω-6:ω-3 was lower in 2013 (3.36 ± 0.79% of total FAs) in comparison to 2012 (3.64 ± 1.14% of total FAs). The difference in FA composition according to production year depended mainly on diet and altitude, whereas animal-related factors, i.e., lactation stage, breed, milk yield, and proportion of primiparous cows in the herd, were not significant [28].

A lower C16:0 and HFA percentage, as well as a higher amount of CLA and C18:1ω-9, most likely results from increased grazing [9]. However, the amount of each FA depends on lipid metabolism in the rumen and the mammary gland and the release of FAs from body reserves during a period of negative energy balance during early lactation since diet (especially type of forage, lipid supplements, forage:concentrate ratio, starch level, and their interactions) can influence metabolic processes in the rumen and the composition of rumen biome [9]. Variations in FA composition can also be explained, in part, by the micro-environmental conditions influencing the quality of feed. The composition of forage and plant lipids is influenced by factors, such as species, growth stage, temperature, and light intensity. Besides, environmental temperature and humidity can affect both milk yield and FA composition since dairy cows are sensitive to climatic variations. The production of the short-chain FAs (C6:0–C8:0) and C18:0 decrease at the higher temperatures and humidity. In this study, 2013 and 2012 were exceptionally hot and wet years, respectively, and there were no differences in the levels C6:0–C10:0 FAs (Table 1).

The results showed no significant seasonal effects (*p* > 0.05) on the percentage of C15:0, C16:1, C20:1ω-9, C20:3ω-6, C21:0, C20:3ω-3, C20:4ω-6, C20:5ω-5, C22:0, and C24:0, although the percentage of total SFAs was lower in summer (68.04 ± 3.20% of total FA) than in winter (70.40 ± 2.06% of total FA, *p* < 0.05). The percentage of MUFAs of total FA was significantly higher (*p* < 0.05) in summer (26.30 ± 3.23%) than in winter (24.20 ± 1.87%) due to increased amounts of C18:1 in summer compared to winter (*p* < 0.05). The percentage of PUFAs in summer (4.11 ± 0.87% of total FA) was higher than in winter (3.43 ± 0.33% of total FA, *p* < 0.05). On average, amounts of SFA decreased in the summer and increased in the winter, whereas the amount of PUFAs increased in summer and decreased in winter. The percentage of ω-3 PUFA was also higher in summer than in winter. Differences in FA percentage related to the season could be explained by differences in the cow’ dietary regimes since during the summer, the cows typically graze on pasture or are fed fresh forage. Replacing grass silages with corn silage increases the concentration of medium-chain saturated FAs (C6:0 to C14:0) and C18:2ω-6 and decreases the concentration of C16:0, C18:0, C18:3ω-3, and C20:5ω-3 in milk fat. Replacing grass silage with corn silage will decrease the amount of total ω-3 PUFA and increase the amount of ω-6 PUFA in milk, leading to an elevated ω-6:ω-3 ratio [12,21,23]. The effect of replacing grass with corn silage on the CLA content of milk is inconclusive. Studies have shown that corn silage-based diets increase the CLA content of milk fat compared with grass-based diets [24,25], while others have found no differences in CLA content of milk fat from cows offered grass or corn silage-based diets [21,26,27]. The present study indicated that the percentage of C18:2ω-6 and CLA was higher during the summer. However, higher environmental temperatures during the summer also affected the FA composition of milk, and milk fat in summer was lower in C16:0 relative to C18:0 than in milk fat from the same cows during winter. Although these changes were related to changes in blood plasma lipids, such observations were confounded by dietary changes. 

As part of this study, the data were analyzed using a clustering approach based on the percentage of FA, production year, season, and corn silage and cereals in the diet. The *δ*^13^C values of casein fraction were also used to characterize the presence of corn silage at a threshold limit of −23.5‰, i.e., samples with *δ*^13^C values higher than −23.5‰ indicated the presence of corn in the diet [16]. Before clustering, the Silhouette curve method was applied to estimate the optimum number of clusters [29]. In this study, the best split was achieved using 14 clusters. The partition around medoids (PAM) method was then applied, which divided the milk samples into 14 clusters (Figure 1). The results showed that the production year and season and the presence of corn silage and cereals in the animal’s diet played a significant role in the clustering. Figure 1 also shows the combined effects of year and season. Production year had a more significant effect on FA composition than the season. Specific FAs showed no global pattern of differentiation and were probably influenced by micro-factors, such as temperature, humidity, feeding regime, lactation stage, breed, and processing techniques.

Multivariate analysis was used to identify the key factors responsible for discriminating between production year and season. In this case, 231 samples consisting of authentic Slovenian milk samples collected in summer and winter 2012 (12 s, 12 w), 2013 (13 s, 13 w), and 2014 (14 s, 14 w) were statistically evaluated. Figure 2 presents the results of the discriminant analysis (DA) as a discriminant function score plot (a) and a discriminant loadings plot (b). In the functional score plot, each group (centroid) was represented by a scatter plot, while in the loadings plot, they appeared as a set of vectors, indicating the degree of association of the corresponding initial variables with the first two discriminant functions (F1 and F2). 

In latter, the degree of distribution of each parameter in the classes was revealed. Except for the 2013 and 2014 winter samples (13 w and 14 w, respectively), good separation between year and season of production was achieved. The most influential parameter for the summer/winter 2012 (12 s/12 w) classes (see top left of biplot a) was C20:4ω-6 (direction of its vector was towards the right part of the biplot), while CLA separated 12 s from 12 w. Winter samples 2013 (13 w) and 2014 (14 w) overlapped, while the separation between summer classes 14 s and 13 s was likely a result of the higher percentage of C17:0iso, C20:1ω-9, C18:3ω-6, C20:3ω-3 (bottom left (a)). A leave-one-out cross-validation (LOOCV) classified 86.9% of the samples correctly. 

#### 2.1.2. Influence of the Geographical Origin on FA Composition 

Fatty acid composition combined with various chemometric tools has been successfully applied in different food commodities to discriminate the geographical origin, including fruits [30], oils [31,32], and also milk and dairy products [26,33,34]. In this study, discriminant analysis (DA) was used to discriminate between milk from Alpine, Mediterranean, Pannonia, and Dinaric regions of Slovenia using the whole dataset, including summer and winter samples from 2012, 2013, and 2014. The results are presented in Figure 3a,b. A comparison of the scatter of data points and vector plots indicated that the Mediterranean group (Figure 2a) was a result of high amounts of C15:0, C17:0, C17:0iso, and C17:0aiso FA. For the Alpine region and Dinaric region groupings (Figure 3c: lower left) C18:3ω-3, CLA, C18:3ω-6, and C20:5ω-3 had the most separating power, while the C12:0, C11:0, C21:0, C18:0, and C18:2ω-6 differentiated the Pannonian group. A cross-validation test showed 64.1% correct classification. The classification rate was the highest (80.8%) for Pannonian milk samples and the lowest for Dinaric samples (20.5%). The more precise and efficient separation between the four regions was obtained when taking into account the season (summer-winter). In this case, the classification rate was >90%. Figure 3c shows the DA plot for 39 milk samples obtained in winter 2013. The analysis generated four well-separated groups. Separation of the regions was also achieved based on FA composition when the data were treated altogether. For F1 (89.0%), C15:0 and its isomers were the main discriminatory parameters, while for F2, the most discriminatory parameters (10.5%) were C18:3ω-6 and CLA. However, cross-validation was low, and only 35.9% of the samples were classified correctly. The highest rate of classification was for the Mediterranean samples (66.7%) and the lowest for the Alpine samples (8.3%). The low classification rate was related to the low number of samples.

### 2.2. Stable Carbon Isotope Composition of FA in Milk 

This study used the stable carbon isotope composition of individual FAs to check the variability in production year and season in 2013 and 2014. The differences measured in the isotopic composition between individual milk FA reflected diet regime, the isotopic fractionation during biosynthesis, metabolic rates, and isotope turnover time. Table 2 presents the stable carbon isotope values of individual FAs in cow milk samples and stable carbon isotope values in bulk freeze-dry milk. The given values were the mean and standard deviation (SD). Results were shown as box plots of *δ*^13^C values of milk fatty acids and bulk milk samples (Figure 4). Data points, exceeding two units of SD, were regarded as outliers, and in the cases where measurements were below the 5% confidence level, the data were discarded. Interestingly, FAs formed during biosynthesis had the lowest *δ*^13^C values due to isotope fractionation.

In all the samples, C15:0 had significantly lower average *δ*^13^C values (−30.3 ± 3.7‰) compared to other FAs. The highest average *δ*^13^C value (−22.6 ± 2.9‰) was observed in C14:0. Higher *δ*^13^C values were found in winter, which reflected the higher amounts of corn in the diet. Although the reported *δ*^13^C values from specific regions, e.g., the Mediterranean region, were influenced by the year of production, significant differences in reported *δ*^13^C values between the summer and the winter season were observed. The fatty acid C18:1ω-9 was an exception with consistently lower values during the summer. It was assumed that this variability was a result of different production practices and the amount of grass and corn forage in a cow’s diet. The C18:1ω-9 in milk from cows fed a C_3_ diet is enriched in ^13^C relative to C18:0, whereas no enrichment occurs when cows’ are fed on a C_4_ diet [17]. In this study, an average enrichment of 1.6‰ was observed in milk sampled during the summer and supported the assumption that a C_3_-based diet could result in the transfer of higher amounts of 18:1 ω-9 directly from the feed. As part of a dietary source, C18:0 can be produced by microbial biohydrogenation of C18:0 as the final product of ruminal biohydrogenation [35]. Microbial biohydrogenation can significantly affect the isotopic values of C18:0, leading to a lower *δ*^13^C values compared to C16:0, which originates in part from dietary lipids and to a significant extent from *de novo* synthesis from acetate, derived mainly from dietary carbohydrates [36]. The Δ^13^C value (*δ*^13^C-18:0−*δ*^13^C-16:0) in the samples were <−3.3‰, which is typical for ruminant fat.

Stable carbon isotope composition of individual FA indicated less variability between season and year than fatty acid composition, which suggested a more distant relationship. There was also no data supporting the correlation between a time-significant percentage of FA and *δ*^13^C values within the same FA. The results of a multivariate analysis of the stable isotope content of FA are presented in Figure 4. The temporal separation between selected groups described 93.8% of the variability. The most important stable isotope parameters that contributed to this separation were: *δ*^13^C-16:0, *δ*^13^C-18:1ω-9*c*, *δ*^13^C-bulk, *δ*^13^C-14:1, and *δ*^13^C-18:0, and a leave-one-out cross-validation (LOOCV) classified 90.9% of the samples correctly.

Figure 4 shows the summer and winter *δ*^13^C values of different FAs in milk for the four studied regions. The *δ*^13^C values of individual fatty acids (season, year, and production region) are given in Appendix A. The *δ*^13^C in FA in milk collected from the Mediterranean region differed significantly (*p* < 0.05) from the other three regions. The differences were more apparent in summer than winter. The Mediterranean region also had more negative *δ*^13^C values, which indicated that cows had a higher amount of grass-based (C_3_ plant) forage in their diet, although the higher temperatures in the Mediterranean region could be a factor. The highest *δ*^13^C values were in FA in milk from the Pannonian region. Besides, the *δ*^13^C-18:1ω-9*c* determined in the Pannonia region differed significantly from Alpine and Dinaric regions. 

### 2.3. Quality of Milk 

Some comparisons can be drawn about the ratios of SFA, MUFA, and PUFA in milk. A healthy diet should contain more PUFA and less of SFA since SFAs are a risk factor for cardiovascular diseases [37]. Alternatively, MUFA and PUFA, except for trans unsaturated fatty acids, reduce the level of low-density lipoprotein (LDL) cholesterol and increase the level of high-density lipoprotein (HDL) cholesterol [38].

The lowest percentage of SFA (avg. 66.97 ± 2.54 % of total FA) was observed in milk produced in summer in the Alpine region, for all three sampling years, while in winter, there were no significant differences between the regions in the SFA percentage of milk sampled in the summer (Table 3). The feeding regime in the Alpine region is based on the managed grazing of pastures during the summer, while in the winter, the cows’ diet consists of a higher amount of cereal concentrates and conserved forages. The feeding of grass conservation products (hay or silage) increases the SFA content of milk fat.

The consensus is that PUFAs, especially ω-3, ω-6, and CLA, have a beneficial effect on human health. Besides, the ω-6:ω-3 ratio, expressed as the concentration of linoleic acid (C18:2ω-6) to α-linolenic (C18:3ω-3) acid, i.e., the most abundant ω-6 and ω-3 FA, as well as atherogenic (AI) and thrombogenic (TI) indices are used to evaluate the nutritional value of milk fat since the milk fat with low AI and TI values could provide protection against coronary heart disease. Further, ω-3 PUFA reduces the risk of cardiovascular disease, hypertension, cancer and has been linked to improved neurological function [5,6,10,13]. The most abundant PUFA in fresh forages is ω-3 (especially α-linolenic acid). In this study, the highest percentage of α-linolenic acid was in the Alpine region in the summer (avg. 0.87 ± 0.38% of total FA) and in the Mediterranean region in winter (avg. 0.71 ± 0.2% of total FA). While the milk from the Pannonian region had the lowest percentage of α-linolenic acid in both summer and winter (avg. 0.62 ± 0.22% of total FA in summer and 0.46 ± 0.07% of total FA in winter). The lower levels of α-linolenic in winter are the result of oxidative losses of PUFA during the wilting of grasses [13]. A correlation also existed between α-linolenic acid and ∑ω-3 FA. The highest percentage of ω-3 FA was present in milk from the Alpine region in summer (avg. 0.97 ± 0.38% of total FA), while in winter, the highest levels of ω-3 FA were observed in the Mediterranean region (avg. 0.83 ± 2.1% of total FA). The lowest percentage of ω-3 FAs was found in milk collected from the Pannonian region in all three production years (avg. 0.76 ± 0.29% of total FA in summer and avg. 0.57 ± 0.09% of total FA in winter). Changes in the ω-3 FA percentage were smaller in the Pannonian region, during both summer and winter compared to the other studied regions. The reason was that the cows’ were fed a diet of corn silage (intensive milk production) and conserve forage throughout the year. Low values of ω-3 FA in low altitude regions of Slovenia (Pannonian and Dinaric regions) were consistent with the literature data for lowlands [39,40]. Since a high ω-6:ω-3 ratio can be detrimental to human health and a factor in cardiovascular disease, cancer, inflammatory and autoimmune disease, an optimum ratio in the human diet of 1:1 ratio is considered, while a ratio <5:1 is recommended [39]. However, a higher ω-6:ω-3 ratio is indicative of higher amounts of cereals (corn, soybean, barley, and oats) in the cows’ diet. The ω-6:ω-3 ratio in Slovenian milk is between 4.2:1 and 2.5:1 and is lower in summer compared to winter. The lowest ω-6:ω-3 ratio was in milk produced in the Mediterranean region through the whole season and did not exceed 2.6:1, which was indicative of a fresh forage-based diet. In the Dinaric and Pannonian regions, the effect of feeding cows corn silage and cereals throughout the year increased the ω-6:ω-3 FA ratio from 3.2:1 to 4.2:1. The data of AI and HI indices were comparable to the study performed by Ferlay et al. [41] determined in the milk of Tarentaise and Montbéliarde cows (3.14 and 3.43, respectively), but they were lower than the data reported by Nantapo et al. [42] in the milk of Friesian, Jersey, and Friesian x Jersey cows with values ranged from 4.08 to 5.13. The highest AI indices in our study were determined in the Mediterranean region (2.89–4.13) and the lowest in the Alpine region (2.31–3.79).

Environmental factors related to year and season together with diet are the most important factors influencing the composition of CLA in milk. In this study, the percentage of CLA ranged from 0.52 ± 0.18% of total FA to 0.73 ± 0.31% of total FA in milk from all four studied regions and increased with dietary intake of 18-carbon PUFAs. Increasing the amount of PUFAs in milk could be achieved by feeding cows’ a diet based on fresh silage. Milk collected in summer in the Alpine region (avg. 0.99 ± 0.28% of total FA) had the highest percentage of CLA, most likely from grazing on pasture. In the winter, there was no significant difference between the Alpine and other regions since the animal’s diet is composed of both corn and grass silage. Because Alpine and Mediterranean milk in summer contained a higher percentage of ω-3 PUFA and CLA, compounds known to have positive effects on human health [40,43], this milk is likely “healthier” than milk produced in other regions in Slovenia. However, despite the benefits of feeding cows fresh grass silage, the overall effects on the FA composition of milk would not improve human health over the long-term [44]. 

## 3. Materials and Methods 

### 3.1. Milk Collection and Storage

During a 3-years study (from 2012 to 2014), in total, 231 raw cow milk samples were collected. Samples were collected twice per year in winter and summer from four geographical regions in Slovenia: Mediterranean (n = 18), Pannonian (n = 90), Dinaric (n = 51), and Alpine (n = 72). Samples were collected from five Slovenian dairies, each supplied by milk from different farms and transported to the laboratory. The geographical characterization of the sampling location: latitude, longitude, altitude, and the number of samples in each location can be found in the Appendix A. Each sample was divided into four subsamples and stored in plastic containers at −20 °C. Sampling data is shown in Table 4.

### 3.2. Chemicals and Standards 

Dichloromethane (CH_2_Cl_2,_ for organic residue analysis), methanol (MeOH, for organic residue analysis), and hexane (C_6_H_14_, for organic residue analysis) were purchased from J. T. Baker B. V. (Deventer, Netherlands). Sodium hydroxide (NaOH, puriss. p.a., ACS reagent, reag. Ph. Eur., ≥98%, pellets) and boron trifluoride-methanol solution (14% in methanol) were obtained from Sigma-Aldrich (St.Louis, MI, USA). Deionized water (18.2 MΩ) was made using a Mili-Q System (Merck Millipore, Watertown, MA, USA). A standard mixture from Supelco (Bellefonte, PA, USA), named Supelco 37 Component FAME Mix in dichloromethane, containing the methyl esters of 37 fatty acids, was used.

The accuracy of the *δ*^13^C determination was checked with the following international reference materials: polyethlene foil IAEA-CH-7 (*δ*^13^C = −32.15 ± 0.04‰) and sucrose IAEA-CH-6 (*δ*^13^C = −10.45 ± 0.03‰) obtained from International Atomic Energy Agency (IAEA, Vienna, Austria), L-glutamic acids USGS40 (*δ*^13^C = −26.39 ± 0.04‰) obtained from U.S. Geology Survey (USGS, Virginia, USA) and laboratory reference materials: casein IAEA-CRP 2013 (*δ*^13^C = −20.3 ± 0.09‰) (IAEA, Vienna, Austria), UreaC (*δ*^13^C = −28.3 ± 0.2‰), and fatty acid methyl nanodecanoate (*δ*^13^C = −30.0 ± 0.1‰) (Restek, Bellefonte, PA, USA).

### 3.3. Analytical Procedures 

#### 3.3.1. Extraction and Esterification of Milk FAs 

Frozen milk samples were thawed (37 °C) and homogenized. Milk fat consists predominantly of triacylglycerides, which were extracted using the in-situ-trans-esterification method without prior extraction of the fat from samples according to the method of Park and Goins (1994). The total lipids fraction was extracted from the milk samples (450 µL) with dichloromethane (300 µL) and 0.5 M sodium hydroxide in methanol (3 mL), purged with nitrogen and heated at 90 °C for 10 min, after which the samples were cooled rapidly. Fatty acid methyl esters (FAMEs) were formed by adding 14% BF_3_-methanol solution (3 mL) and heating the sample at 90 °C for a further 10 min. After cooling, the fatty acids methyl esters (FAMEs) were extracted with hexane (1.5 mL) and transferred to a GC vial and stored at −20 °C.

#### 3.3.2. Gas Chromatography of Fatty Acids 

The characterization of FAMEs was performed using Agilent 6890N (Network GC System, Agilent Technology, Santa Clara, CA, USA) gas chromatography with FID detector (GC-FID). The separation was achieved on an Omegawax 320, 30 m × 0.32 mm, × 0.25 µm, on a capillary column (Supelco, Bellefonte, PA, USA). The temperature program was as follows: 50 °C (held 2 min) to 220 °C at 4 °C/min (held 20 min). The carrier gas was helium in constant flow mode (1 mL/min). The injector was set to 260 °C and the detector to 280 °C. The individual fatty acids were identified and quantified by comparing their retention times with those of a standard Supelco 37 component FAME Mix (Supelco) and expressed in weight percent of total fatty acids. Procedural blanks were analyzed with each set of samples. Standard Supelco 37 component FAME mix was analyzed after every ten samples to verify the stability of the analytical system. Method precision based on measurements of replicate (n = 2) real samples was 5%. Results were expressed as the percentage of each fatty acid with respect to the total fatty acids [45]. This percentage was calculated using the peak area of the samples corrected with the respective correction factors, according to Christie and Han [46].

#### 3.3.3. Elemental Analysis-Isotope Ratio Mass Spectrometry (EA-IRMS)

The stable carbon isotope ratio measurements were reported in *δ*—notation expressed in per mile (‰) relative to the Vienna-Pee Dee Belemnite (V-PDB) standard following Equation (1) [47]:(1)δ(i/jE)=δi/jE=RPi/j−i/jRRefi/jRRef
where superscripts *i* and *j* denote the highest and the lowest atomic mass number of element E, and *R_P_* and *R_Ref_* is the ratio between the heavier and the lighter isotopes (in case of carbon, ^13^C/^12^C) in the sample (*P*) and reference material (*Ref*). 

To determine the bulk carbon isotope composition, approximately 1 mg of freeze-dried milk was weighed into a tin capsule, closed with tweezers, and placed into the automatic sampler of the elemental analyzer. All analyses were performed separately on a Europa Scientific 20–20 continuous flow mass spectrometer with an ANCA-SL solid–liquid preparation module (Sercon, Crewe, UK). Analyses were normalized against international standards: IAEA-CH-7 and IAEA-CH-6 with *δ*^13^C values of −32.15 ± 0.04‰ and −10.45 ± 0.04‰, respectively. For independent control, laboratory reference material IAEA-CRP 2013 with *δ*^13^C value −20.3 ± 0.09‰ and UreaC with *δ*^13^C value −28.3 ± 0.2‰ were used. The analytical precision, expressed as the standard deviation of control material, was ± 0.2‰.

The ^13^C/^12^C ratio of synthetic fatty acid standard C19:0 (methyl nanodecanoate, RESTEK) was determined using a Europa Scientific 20–20 continuous flow mass spectrometer with an ANCA-SL solid–liquid preparation module (Sercon, Crewe, UK). Results were further normalized against two international reference standards: IAEA-CH-7 and IAEA-CH-6. For independent control, the international reference material USGS40 with *δ*^13^C values of −26.39 ± 0.04‰ was used. Analytical precision, expressed as the standard deviation of the control materials, was ± 0.2‰.

#### 3.3.4. Compound Specific Isotope Analysis of δ^13^C in Fatty Acids 

The Isotopic compositions of individual FA were determined using an Agilent 6890N GC-C system coupled to an IsoPrime GV IRMS (GV Instruments, Manchester, UK)). The separation was achieved using a DB-1MS (60 m × 0.32 mm × 0.25 µm) capillary column (Agilent J&W, USA). The temperature program was as follows: 120 °C (held 1 min) to 300 °C (held 20 min) at 3 °C/min. The carrier gas was helium in the constant flow mode (1 mL/min). The injector temperature was 260 °C, and the oxidation reactor (Cu/O) in the 6890N GC/C system was 900 °C. Peak identification of individual fatty acids was performed by comparing their retention times with those in the standard Supelco 37 component FAME Mix, which was analyzed in the measurement sequence. Instrument stability was ≤0.08‰. C19:0 FAME (methyl nanodecanoate) with *δ*^13^C values of −30 ± 0.1‰ (prior determined on EA-IRMS) was used as a laboratory reference material. For data normalization, a single-point normalization method was used [48]. The precision of measurements expressed as the standard deviation of the laboratory reference material was 0.3 to 0.5‰. 

### 3.4. Statistical Evaluation 

For statistical evaluation, FAs were considered individually and grouped in classes as follows: saturated fatty acids (SFA), unsaturated fatty acids (UFA), MUFA, ω-3 PUFA, ω-6 PUFA, and CLA. Statistical calculations and multivariate analysis were carried out using the XLSTAT software package (2020.1.3, Addinsoft, New York, NY, USA). The statistical evaluation was performed using R, where non-parametric tests were used, i.e., Kruskal–Wallis followed by a post hoc test. In the case of pairwise comparison, a Mann–Whitney test was used. Basic statistics included mean values (median and arithmetic mean), standard deviation (SD), minimum and maximum, while for multivariate analysis, discriminant analysis (DA) was used.

## 4. Conclusions

Slovenian milk samples had been classified according to the year, season, and region of production based on the determination of the percentage and stable isotope composition of individual fatty acids. It was found that the FA composition differed significantly, depending more on year and season of production than geographical region. A leave-one-out cross-validation (LOOCV) classified 86.9% of the samples correctly. For geographical origin determination, it was found that the Mediterranean group was separated from other regions based on a high percentage of C15:0 and C17:0 and their isomers. The cross-validation test showed 64.1% correct classification with the highest rate of classification for the Pannonian (80.8%) samples. Although the classification rate according to the year/season of production improved when stable isotope composition of individual FA was included, describing 93.8% of variability, it was found that *δ*^13^C in FA could be more useful for studying metabolic transformation processes in ruminants. It appeared that the natural *δ*^13^C differences in FA depended on both—the diet type and the precursor FA. The “healthiest” milk for human consumption, based on FA composition, was produced in the Alpine and Mediterranean regions. The quality of milk was also affected by summer/winter season production. The results gave proof of the concept that FA composition could help to discriminate the geographical origin of milk and thus identify products of better quality. Finally, to confirm the validity of the markers, future studies should focus on how factors, such as breed and age, lactation period, production level, the farming system, influence the FA and stable isotope composition of milk. 

## Figures and Tables

**Figure 1 molecules-25-02892-f001:**
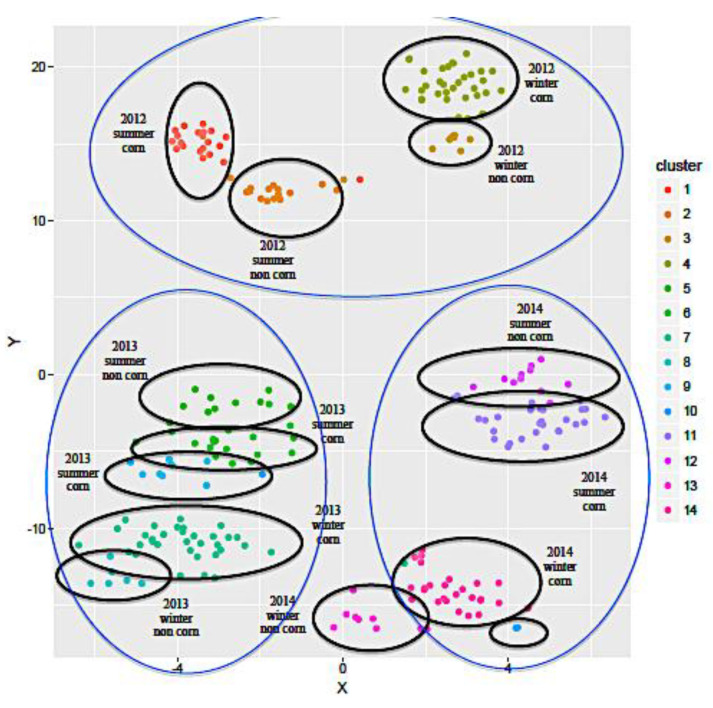
Cluster analysis of the percentage of fatty acid (FA) in Slovenian milk.

**Figure 2 molecules-25-02892-f002:**
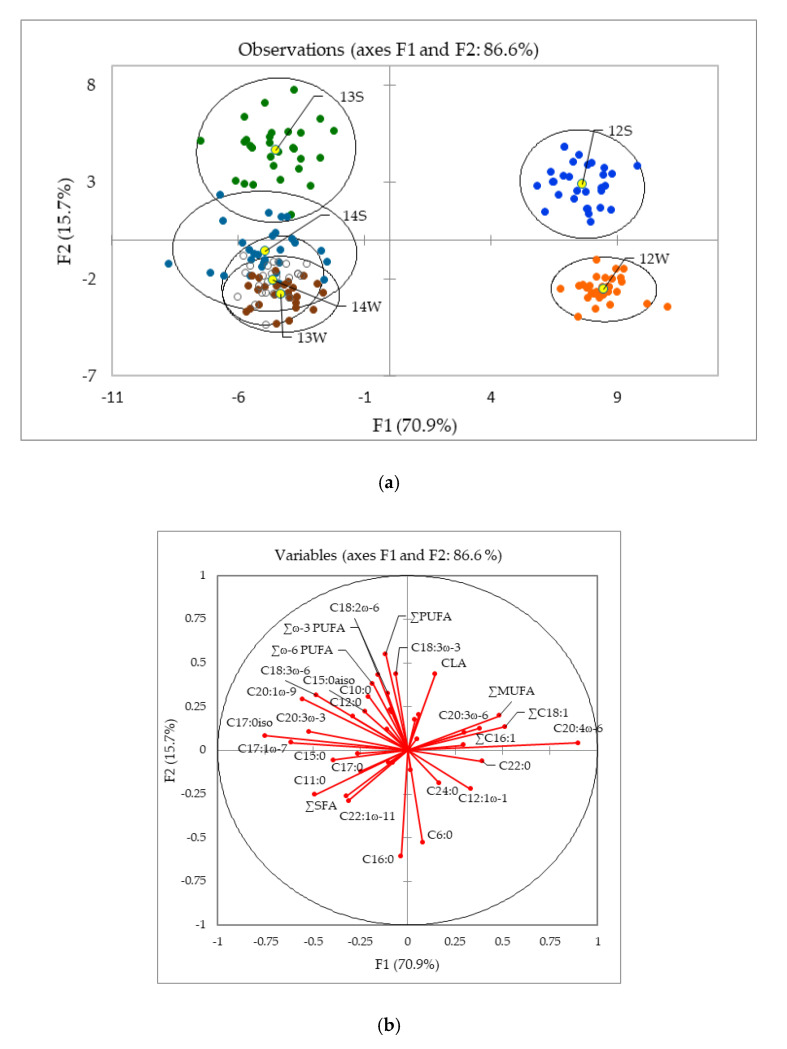
Linear discriminant analysis of FA in milk according to year and season of production: (**a**) discriminant function score plot of 231 authentic Slovenian milk samples from summer/winter in 2012, 2013, and 2014 (12 s/12 w; 13 s/13 w; 14 s/14 w); (**b**) discriminant loading plot of the correlations between the initial variables and the discriminant functions.

**Figure 3 molecules-25-02892-f003:**
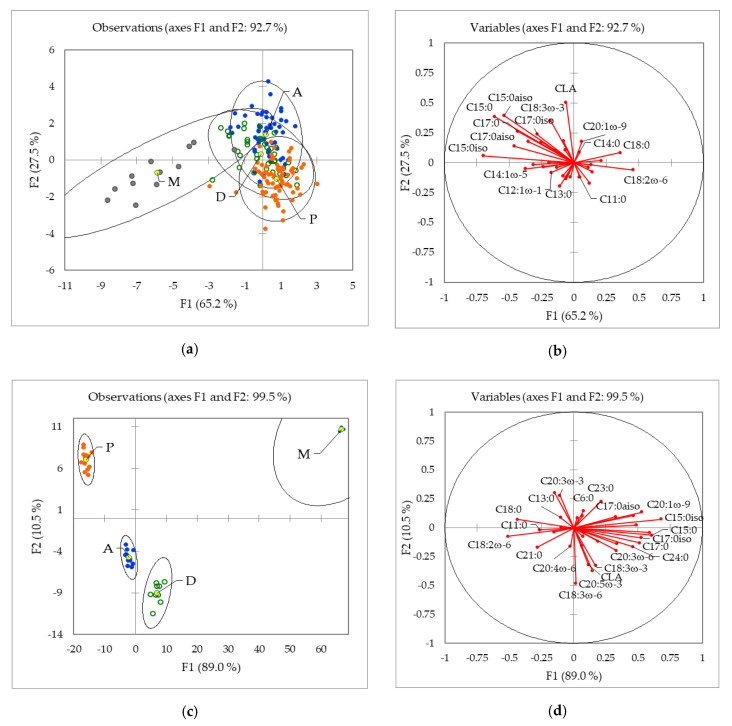
Linear discriminant analysis of FA composition according to the production region: M-Mediterranean, A-Alpine, D-Dinaric, and P-Pannonian; (**a**) discriminant function score plot of 231 authentic Slovenian milk samples from summer/winter: 2012, 2013, and 2014; (**c**) discriminant function score plot of 39 Slovenian milk samples from four regions in winter 2013; (**b**,**d**) discriminant loading plot, showing correlations between initial variables and discriminant functions.

**Figure 4 molecules-25-02892-f004:**
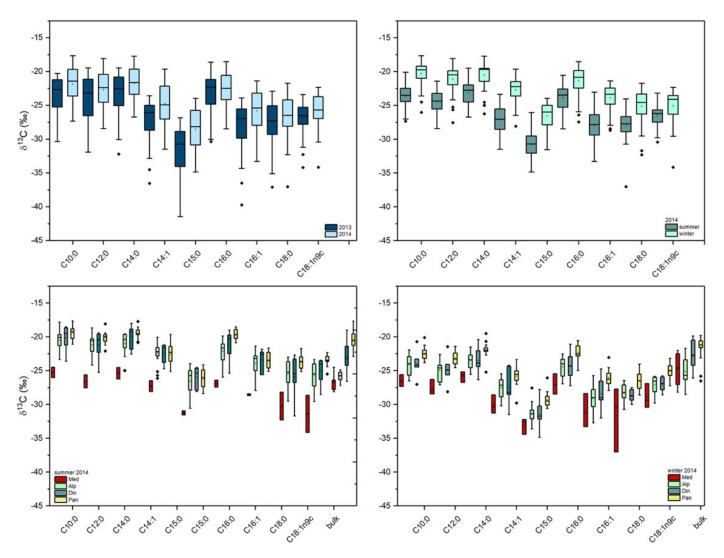
Box plots of *δ*^13^C values of milk fatty acids and bulk milk samples according to the year (2013, 2014), season (summer, winter in 2014), and region of production in summer and winter 2014: Med-Mediterranean, Alp-Alpine, Din-Dinaric, Pan-Pannonian.

**Table 1 molecules-25-02892-t001:** The percentage of fatty acid (FA) in Slovenian milk according to the year (2012, 2013, and 2014), season (winter (12 w, 13 w, 14 w), summer (12 s, 13 s, 14 s), and type of feed. Influence of the feed type on milk FA composition [9,12,13,28].

		2012	2013	2014	Winter	Summer	Corn	Non-Corn	*p*-Value ^1^	
FA (% of Total FA)	Feed Type	Mean ± SD							Year	Season
**SFA**										
C6:0	Corn	1.25 ± 0.26	1.20 ± 0.46	1.30 ± 0.44	1.44 ± 0.32	1.05 ± 0.37	1.30 ± 0.39	1.12 ± 0.40	*	***
C8:0	Corn, concentrates	1.22 ± 0.08	1.36 ± 0.41	1.22 ± 0.34	1.28 ± 0.26	1.25 ± 0.37	1.29 ± 0.33	1.20 ± 0.28	***	*
C10:0	Corn, concentrates	3.03 ± 0.25	3.57 ± 1.02	3.18 ± 0.82	3.28 ± 0.62	3.23 ± 0.95	3.33 ± 0.81	3.07 ± 0.74	**	*
C11:0	Corn, concentrates	0.04 ± 0.03	0.06 ± 0.03	0.08 ± 0.02	0.06 ± 0.02	0.05 ± 0.04	0.06 ± 0.03	0.05 ± 0.03	***	***
C12:0	Corn, concentrates	3.57 ± 0.33	4.10 ± 0.80	3.90 ± 0.82	3.97 ± 0.61	3.74 ± 0.80	3.94 ± 0.67	3.65 ± 0.79	NS	***
C13:0		0.11 ± 0.01	0.11 ± 0.03	0.11 ± 0.03	0.12 ± 0.02	0.10 ± 0.03	0.11 ± 0.03	0.10 ± 0.02	NS	***
C14:0	Corn	12.0 ± 0.65	12.8 ± 1.86	12.9 ± 1.26	12.9 ± 0.94	12.3 ± 1.70	12.6 ± 1.41	12.5 ± 1.39	NS	***
C15:0iso	Hay, grass silage	0.32 ± 0.05	0.34 ± 0.13	0.34 ± 0.05	0.32 ± 0.05	0.35 ± 0.12	0.32 ± 0.09	0.37 ± 0.07	***	***
C15:0aiso	Hay, grass silage	0.51 ± 0.08	0.53 ± 0.09	0.56 ± 0.09	0.50 ± 0.07	0.56 ± 0.10	0.50 ± 0.07	0.61 ± 0.11	NS	***
C15:0	Hay, grass silage	1.17 ± 0.11	1.24 ± 0.13	1.33 ± 0.15	1.24 ± 0.14	1.25 ± 0.16	1.21 ± 0.12	1.34 ± 0.16	NS	NS
C16:0	Calcium salts of palm oil, canola and soy, corn	32.60 ± 2.13	32.7 ± 2.48	33.2 ± 2.11	34.1 ± 1.73	31.6 ± 2.01	32.9 ± 1.86	32.4 ± 3.04	***	***
C17:0iso	Hay, grass silage	0.37 ± 0.07	0.50 ± 0.06	0.53 ± 0.06	0.44 ± 0.10	0.49 ± 0.09	0.45 ± 0.09	0.51 ± 0.10	***	***
C17:0aiso	Hay, grass silage	0.47 ± 0.05	0.44 ± 0.06	0.48 ± 0.07	0.44 ± 0.05	0.48 ± 0.07	0.45 ± 0.05	0.51 ± 0.08	***	***
C17:0	Hay, grass silage	0.61 ± 0.08	0.65 ± 0.11	0.69 ± 0.15	0.63 ± 0.10	0.67 ± 0.14	0.61 ± 0.09	0.73 ± 0.15	NS	***
C18:0	Fresh herbage, grass silage, concentrates	10.2 ± 1.01	9.87 ± 1.17	9.99 ± 1.66	9.41 ± 0.90	10.6 ± 1.36	9.97 ± 1.06	10.2 ± 1.81	***	***
C20:0		0.18 ± 0.02	0.19 ± 0.15	0.14 ± 0.08	0.16 ± 0.05	0.18 ± 0.13	0.17 ± 0.11	0.17 ± 0.07	NS	***
C21:0		0.03 ± 0.01	0.05 ± 0.07	0.02 ± 0.02	0.04 ± 0.03	0.03 ± 0.05	0.03 ± 0.05	0.03 ± 0.03	***	NS
C22:0		0.07 ± 0.02	0.04 ± 0.04	0.03 ± 0.03	0.05 ± 0.03	0.05 ± 0.04	0.05 ± 0.03	0.05 ± 0.03	***	NS
C23:0		0.03 ± 0.01	0.04 ± 0.05	0.03 ± 0.03	0.04 ± 0.04	0.03 ± 0.02	0.03 ± 0.04	0.04 ± 0.03	***	***
C24:0		0.05 ± 0.01	0.04 ± 0.04	0.04 ± 0.03	0.05 ± 0.02	0.04 ± 0.03	0.04 ± 0.03	0.05 ± 0.03	***	NS
∑SFA	Conserved forage and concentrate rich diet	67.76 ± 2.49	69.90 ± 2.99	70.10 ± 2.74	70.40 ± 2.06	68.04 ± 3.20	69.43 ± 2.61	68.71 ± 3.60	***	***
**MUFA**	Rapeseed, seeds oils, fresh grass, non-conserved forage									
C12:1		0.12 ± 0.03	0.10 ± 0.03	0.09 ± 0.03	0.12 ± 0.03	0.09 ± 0.03	0.10 ± 0.03	0.10 ± 0.04	***	***
C14:1		1.04 ± 0.11	1.29 ± 1.41	1.17 ± 0.25	1.18 ± 0.18	1.15 ± 1.17	1.19 ± 0.95	1.12 ± 0.29	NS	***
C16:1		2.03 ± 0.11	1.85 ± 0.20	1.94 ± 0.17	1.92 ± 0.14	1.96 ± 0.21	1.95 ± 0.18	1.92 ± 0.18	***	NS
C17:1		0.28 ± 0.03	0.34 ± 0.05	0.37 ± 0.06	0.31 ± 0.06	0.34 ± 0.07	0.32 ± 0.05	0.36 ± 0.07	***	***
C18:1ω-9*c*/9*t*	Grass, rapeseeds	23.7 ± 2.17	20.12 ± 2.8	20.7 ± 2.56	20.5 ± 2.01	22.6 ± 3.38	21.4 ± 2.76	21.9 ± 3.42	***	***
C20:1ω-9		0.04 ± 0.01	0.25 ± 0.17	0.14 ± 0.08	0.12 ± 0.09	0.17 ± 0.17	0.14 ± 0.13	0.16 ± 0.16	***	NS
C22:1ω-9		0.01 ± 0.01	0.02 ± 0.02	0.04 ± 0.03	0.02 ± 0.02	0.02 ± 0.02	0.02 ± 0.02	0.02 ± 0.02	NS	***
∑MUFA		27.26 ± 2.10	23.97 ± 2.83	24.42 ± 2.38	24.20 ± 1.87	26.30 ± 3.23	25.10 ± 2.71	25.54 ± 3.17	***	***
**PUFA**										
CLA (*c*-9, *t*-11 C18:2)	Grass silage, fish oil, rapeseed, linseed	0.68 ± 0.28	0.61 ± 0.26	0.55 ± 0.20	0.51 ± 0.12	0.72 ± 0.31	0.55 ± 0.19	0.78 ± 0.31	***	***
**ω-6 PUFA**	Corn, hay, straw,									
C18:2ω-6*c*	Grassing, red clover silage, corn	2.00 ± 0.22	2.09 ± 0.32	1.98 ± 0.28	1.94 ± 0.23	2.11 ± 0.30	2.05 ± 0.24	1.95 ± 0.35	NS	***
C18:3ω-6		0.11 ± 0.02	0.43 ± 0.29	0.23 ± 0.12	0.20 ± 0.12	0.32 ± 0.30	0.24 ± 0.21	0.31 ± 0.28	***	***
C20:3ω-6	Fish oil	0.08 ± 0.01	0.06 ± 0.04	0.06 ± 0.04	0.06 ± 0.04	0.06 ± 0.04	0.07 ± 0.04	0.06 ± 0.03	***	NS
C20:4ω-6	Fish oil	0.12 ± 0.02	0.03 ± 0.02	0.03 ± 0.03	0.06 ± 0.05	0.06 ± 0.05	0.06 ± 0.05	0.06 ± 0.05	***	NS
∑ω-6 PUFA		2.31 ± 0.24	2.61 ± 0.58	2.30 ± 0.33	2.27 ± 0.26	2.55 ± 0.53	2.41 ± 0.39	2.38 ± 0.54	***	***
**ω-3 PUFA**	Grassing,									
C18:3ω-3	Grassing, red clover silage, legume silage, cereals, oil seeds	0.62 ± 0.20	0.69 ± 0.33	0.58 ± 0.20	0.54 ± 0.14	0.72 ± 0.30	0.57 ± 0.23	0.78 ± 0.25	NS	***
C20:3ω-3	Fish oil	0.02 ± 0.01	0.10 ± 0.09	0.09 ± 0.03	0.06 ± 0.05	0.07 ± 0.08	0.07 ± 0.07	0.05 ± 0.03	***	NS
C20:5ω-3	Fish oil, linseed	0.06 ± 0.02	0.05 ± 0.05	0.04 ± 0.03	0.05 ± 0.02	0.05 ± 0.06	0.05 ± 0.04	0.07 ± 0.06	***	NS
∑ω-3 PUFA		0.70 ± 0.22	0.84 ± 0.34	0.71 ± 0.21	0.65 ± 0.15	0.85 ± 0.32	0.69 ± 0.25	0.90 ± 0.25	***	***
∑PUFA		3.69 ± 0.50	4.05 ± 0.98	3.55 ± 0.56	3.43 ± 0.33	4.11 ± 0.87	3.65 ± 0.65	4.06 ± 0.86	***	***
ω-6:ω-3		3.64 ± 1.14	3.36 ± 0.79	3.80 ± 3.18	3.71 ± 1.19	3.48 ± 2.58	3.87 ± 2.14	2.87 ± 1.35	NS	***

SFA = saturated fatty acids; MUFA = monosaturated fatty acids; PUFA = polystaurated fatty acids; ^1^ NS = not significant; * *p* < 0.05; ** *p* < 0.01; *** *p* < 0.001.

**Table 2 molecules-25-02892-t002:** *δ*^13^C values (in ‰) in FAs in Slovenian milk according to the year, season, and feeding regime in 2013 and 2014. The source of FAs in milk, which influences its isotopic composition, is summarized by Ehtesham et al. [25], Richter et al. [18], Hanus et al. [9].

	*δ*^13^C (‰) Mean ± SD							
FA	Source of FA	2013	2014	Winter	Summer	Corn	Non-Corn	Average
**C10:0**	metabolism	−23.9 ± 2.7	−22.1 ± 2.5	−21.3 ± 2.0	−24.7 ± 2.3	−22.2 ± 2.4	−26.2 ± 1.8	−22.9 ± 2.8
**C12:0**	metabolism	−24.6 ± 3.2	−22.8 ± 2.6	−21.8 ± 2.2	−25.7 ± 2.5	−22.8 ± 2.6	−27.3 ± 2.0	−23.6 ± 3.0
**C14:1**	biosynthesis	−27.4 ± 3.2	−24.9 ± 3.0	−24.3 ± 2.5	−28.0 ± 3.0	−25.2 ± 2.9	−29.9 ± 2.2	−26.0 ± 3.3
**C14:0**	metabolism	−23.5 ± 3.0	−21.9 ± 2.6	−21.2 ± 2.0	−24.3 ± 2.8	−21.9 ± 2.4	−26.2 ± 2.2	−22.6 ± 2.9
**C15:0**	microbial biosynthesis	−32.1 ± 3.7	−28.8 ± 2.9	−28.2 ± 2.6	−32.6 ± 3.4	−29.5 ± 3.4	−33.9 ± 2.7	−30.3 ± 3.7
**C16:1**	biosynthesis	−28.5 ± 3.5	−26.0 ± 3.0	−25.2 ± 2.3	−29.4 ± 3.3	−26.3 ± 3.0	−31.3 ± 2.6	−27.2 ± 3.5
**C16:0**	diet and metabolism	−23.6 ± 2.9	−22.6 ± 2.5	−21.8 ± 2.0	−24.5 ± 2.7	−22.2 ± 2.2	−26.7 ± 1.7	−23.0 ± 2.7
**C18:1ω-9c**	biosynthesis	−27.2 ± 2.5	−25.9 ± 2.2	−25.6 ± 2.1	−27.5 ± 2.5	−25.8 ± 1.9	−29.8 ± 2.1	−26.5 ± 2.5
**C18:0**	biosynthesis and diet	−27.8 ± 3.2	−26.6 ± 2.8	−25.6 ± 2.3	−29.0 ± 2.7	−26.4 ± 2.5	−31.0 ± 2.6	−27.2 ± 3.0
**Bulk**	diet	−21.4 ± 1.9	−21.5 ± 2.2	−20.6 ± 1.7	−22.5 ± 1.9	−20.7 ± 1.4	−24.8 ± 0.9	−21.4 ± 2.0

**Table 3 molecules-25-02892-t003:** Percentage of FA groups in Slovenian milk: 2012 and 2014, based on geographical origin and season.

Fatty Acids	Year	Season	Alpine	Mediterranean	Pannonian	Dinaric
(% of Total FA)			Mean ± SD			
Saturated FA (SFA)	2012	summer	64.15 ± 1.62	69.57 ± 3.30	66.33 ± 0.74	66.74 ± 2.14
		winter	69.57 ± 0.82	71.64 ± 3.10	68.89 ± 0.66	69.52 ± 0.53
		average	66.81 ± 3.09	70.60 ± 3.08	67.69 ± 1.47	68.12 ± 2.05
	2013	summer	68.78 ± 1.60	68.19 ± 7.58	70.20 ± 5.04	70.63 ± 0.71
		winter	70.50 ± 0.75	70.39 ± 3.68	69.63 ± 1.99	70.08 ± 2.46
		average	69.64 ± 1.51	69.29 ± 5.46	69.91 ± 3.77	70.34 ± 1.82
	2014	summer	68.08 ± 1.33	69.55 ± 4.65	68.42 ± 1.22	68.62 ± 3.48
		winter	72.45 ± 1.02	73.23 ± 4.25	70.56 ± 2.76	72.16 ± 0.79
		average	70.26 ± 2.51	71.39 ± 4.47	69.42 ± 2.31	70.49 ± 2.99
Monounsaturated FA (MUFA)	2012	summer	29.93 ± 1.28	24.95 ± 3.35	28.73 ± 0.62	27.96 ± 2.00
		winter	25.79 ± 0.86	23.59 ± 3.26	26.56 ± 0.52	25.94 ± 0.51
		average	27.86 ± 2.37	24.27 ± 3.05	27.57 ± 1.23	26.89 ± 1.72
	2013	summer	24.98 ± 1.38	26.40 ± 7.03	24.36 ± 2.00	23.93 ± 0.86
		winter	23.73 ± 0.84	24.16 ± 3.09	24.16 ± 0.53	23.26 ± 1.13
		average	24.36 ± 1.29	25.28 ± 5.01	24.25 ± 1.42	23.57 ± 1.04
	2014	summer	26.31 ± 0.92	25.53 ± 3.66	26.74 ± 1.14	25.13 ± 2.27
		winter	22.20 ± 1.13	21.59 ± 3.56	23.15 ± 1.37	22.64 ± 0.57
		average	24.25 ± 2.33	23.56 ± 3.89	25.07 ± 2.20	23.81 ± 2.01
Conjugated linoleic acid (C18:2*c*9,*t*11)	2012	summer	1.20 ± 0.30	0.67 ± 0.12	0.62 ± 0.12	0.72 ± 0.20
		winter	0.61 ± 0.09	0.59 ± 0.10	0.48 ± 0.06	0.55 ± 0.10
		average	0.90 ± 0.37	0.63 ± 0.11	0.54 ± 0.11	0.63 ± 0.17
	2013	summer	0.89 ± 0.24	0.67 ± 0.61	0.63 ± 0.34	0.54 ± 0.12
		winter	0.62 ± 0.15	0.52 ± 0.23	0.44 ± 0.07	0.54 ± 0.15
		average	0.75 ± 0.24	0.60 ± 0.42	0.54 ± 0.26	0.54 ± 0.13
	2014	summer	0.86 ± 0.18	0.51 ± 0.31	0.50 ± 0.14	0.58 ± 0.21
		winter	0.47 ± 0.09	0.53 ± 0.20	0.44 ± 0.08	0.43 ± 0.06
		average	0.67 ± 0.24	0.52 ± 0.23	0.47 ± 0.12	0.51 ± 0.16
α-linolenic acid (C18:3ω-3)	2012	summer	0.84 ± 0.14	0.98 ± 0.34	0.52 ± 0.07	0.70 ± 0.16
		winter	0.58 ± 0.09	0.76 ± 0.33	0.45 ± 0.06	0.57 ± 0.16
		average	0.71 ± 0.18	0.87 ± 0.32	0.48 ± 0.07	0.63 ± 0.17
	2013	summer	1.08 ± 0.54	0.46 ± 0.07	0.78 ± 0.30	0.71 ± 0.21
		winter	0.61 ± 0.12	0.56 ± 0.14	0.50 ± 0.05	0.56 ± 0.11
		average	0.84 ± 0.45	0.51 ± 0.11	0.64 ± 0.26	0.63 ± 0.18
	2014	summer	0.69 ± 0.26	0.89 ± 0.24	0.54 ± 0.13	0.66 ± 0.21
		winter	0.60 ± 0.13	0.81 ± 0.08	0.45 ± 0.09	0.45 ± 0.17
		average	0.64 ± 0.21	0.85 ± 0.17	0.50 ± 0.12	0.55 ± 0.22
Eicosapentaenoic acid (C20:5ω-3)	2012	summer	0.08 ± 0.01	0.11 ± 0.02	0.06 ± 0.01	0.07 ± 0.02
		winter	0.06 ± 0.01	0.08 ± 0.02	0.05 ± 0.02	0.06 ± 0.02
		average	0.07 ± 0.01	0.09 ± 0.03	0.05 ± 0.01	0.07 ± 0.02
	2013	summer	0.02 ± 0.03	0.16 ± 0.26	0.06 ± 0.10	0.02 ± 0.02
		winter	0.05 ± 0.04	0.04 ± 0.01	0.03 ± 0.02	0.05 ± 0.02
		average	0.04 ± 0.04	0.10 ± 0.17	0.05 ± 0.07	0.04 ± 0.02
	2014	summer	0.04 ± 0.02	0.03 ± 0.03	0.03 ± 0.04	0.03 ± 0.02
		winter	0.04 ± 0.02	0.07 ± 0.01	0.04 ± 0.01	0.07 ± 0.02
		average	0.04 ± 0.02	0.05 ± 0.03	0.03 ± 0.03	0.05 ± 0.03
Polyunsaturated FA (PUFA)	2012	summer	4.59 ± 0.46	3.71 ± 0.18	3.68 ± 0.19	3.90 ± 0.27
		winter	3.40 ± 0.17	3.43 ± 0.25	3.39 ± 0.21	3.32 ± 0.26
		average	4.00 ± 0.70	3.57 ± 0.25	3.52 ± 0.25	3.59 ± 0.39
	2013	summer	4.98 ± 1.04	3.76 ± 1.38	4.58 ± 1.36	4.06 ± 0.61
		winter	3.81 ± 0.38	3.24 ± 0.44	3.44 ± 0.19	3.66 ± 0.29
		average	4.39 ± 0.97	3.50 ± 0.96	4.01 ± 1.12	3.85 ± 0.50
	2014	summer	4.23 ± 0.66	3.64 ± 0.73	3.53 ± 0.39	3.69 ± 0.67
		winter	3.41 ± 0.31	3.21 ± 0.72	3.32 ± 0.31	3.20 ± 0.37
		average	3.82 ± 0.66	3.42 ± 0.69	3.43 ± 0.37	3.43 ± 0.57
ω-3 PUFA	2012	summer	0.94 ± 0.16	1.12 ± 0.37	0.59 ± 0.08	0.79 ± 0.19
		winter	0.66 ± 0.10	0.86 ± 0.35	0.51 ± 0.06	0.64 ± 0.18
		average	0.80 ± 0.19	0.99 ± 0.35	0.55 ± 0.08	0.71 ± 0.20
	2013	summer	1.17 ± 0.54	0.65 ± 0.28	1.00 ± 0.37	0.83 ± 0.16
		winter	0.75 ± 0.14	0.70 ± 0.14	0.63 ± 0.06	0.68 ± 0.12
		average	0.96 ± 0.44	0.67 ± 0.20	0.82 ± 0.32	0.75 ± 0.16
	2014	summer	0.81 ± 0.28	1.01 ± 0.28	0.67 ± 0.16	0.75 ± 0.22
		winter	0.73 ± 0.16	0.95 ± 0.07	0.58 ± 0.10	0.61 ± 0.17
		average	0.77 ± 0.22	0.98 ± 0.18	0.63 ± 0.14	0.68 ± 0.20
ω-6 PUFA	2012	summer	2.45 ± 0.13	1.92 ± 0.16	2.48 ± 0.15	2.38 ± 0.17
		winter	2.14 ± 0.15	1.98 ± 0.08	2.39 ± 0.23	2.14 ± 0.25
		average	2.29 ± 0.21	1.95 ± 0.12	2.43 ± 0.20	2.25 ± 0.24
	2013	summer	2.92 ± 0.70	2.43 ± 1.23	2.94 ± 0.85	2.70 ± 0.51
		winter	2.44 ± 0.18	2.03 ± 0.22	2.36 ± 0.14	2.43 ± 0.21
		average	2.68 ± 0.55	2.23 ± 0.82	2.65 ± 0.67	2.56 ± 0.39
	2014	summer	2.56 ± 0.34	2.12 ± 0.18	2.36 ± 0.30	2.35 ± 0.33
		winter	2.20 ± 0.16	1.72 ± 0.55	2.29 ± 0.22	2.16 ± 0.34
		average	2.38 ± 0.32	1.92 ± 0.43	2.33 ± 0.26	2.25 ± 0.34
ω-6:ω-3 PUFA	2012	summer	2.69 ± 0.55	1.90 ± 0.85	4.26 ± 0.72	3.19 ± 0.94
		winter	3.32 ± 0.61	2.58 ± 1.03	4.79 ± 0.94	3.60 ± 1.08
		average	3.01 ± 0.66	2.24 ± 0.92	4.54 ± 0.87	3.41 ± 1.01
	2013	summer	2.86 ± 1.05	3.64 ± 0.42	3.15 ± 0.86	3.34 ± 0.73
		winter	3.35 ± 0.52	3.01 ± 0.78	3.79 ± 0.39	3.69 ± 0.91
		average	3.11 ± 0.84	3.32 ± 0.66	3.47 ± 0.73	3.52 ± 0.82
	2014	summer	3.01 ± 0.58	2.19 ± 0.48	3.69 ± 0.89	3.30 ± 0.74
		winter	3.13 ± 0.61	1.83 ± 0.67	4.06 ± 0.72	3.32 ± 0.86
		average	3.07 ± 0.59	2.01 ± 0.56	3.86 ± 0.82	3.31 ± 0.69
AI	2012	summer	2.31 ± 0.20	3.25 ± 0.53	2.58 ± 0.10	2.63 ± 0.33
		winter	3.06 ± 0.15	3.51 ± 0.60	2.92 ± 0.10	3.06 ± 0.11
		average	2.69 ± 0.42	3.38 ± 0.52	2.76 ± 0.20	2.86 ± 0.32
	2013	summer	3.03 ± 0.28	2.89 ± 1.82	3.15 ± 0.46	3.33 ± 0.20
		winter	3.30 ± 0.19	3.34 ± 0.62	3.79 ± 0.39	3.71 ± 0.21
		average	3.16 ± 0.27	3.13 ± 1.24	3.47 ± 0.73	3.35 ± 0.18
	2014	summer	2.81 ± 0.27	2.92 ± 0.75	2.84 ± 0.20	3.07 ± 0.43
		winter	3.79 ± 0.25	4.13 ± 1.05	3.50 ± 0.31	3.75 ± 0.19
		average	3.30 ± 0.59	3.53 ± 1.05	3.15 ± 0.42	3.43 ± 0.47
TI	2012	summer	2.69 ± 0.23	3.19 ± 0.23	3.05 ± 0.11	3.03 ± 0.28
		winter	3.47 ± 0.11	3.62 ± 0.28	3.43 ± 0.11	3.46 ± 0.11
		average	3.08 ± 0.43	3.41 ± 0.33	3.25 ± 0.22	3.26 ± 0.30
	2013	summer	3.09 ± 0.34	3.18 ± 1.13	3.19 ± 0.42	3.41 ± 0.20
		winter	3.67 ± 0.14	3.68 ± 0.58	3.64 ± 0.12	3.71 ± 0.21
		average	3.38 ± 0.40	3.43 ± 0.85	3.42 ± 0.38	3.57 ± 0.25
	2014	summer	3.25 ± 0.34	3.35 ± 0.71	3.33 ± 0.20	3.46 ± 0.44
		winter	3.82 ± 0.14	3.88 ± 0.79	3.80 ± 0.16	3.95 ± 0.25
		average	3.54 ± 0.39	3.61 ± 0.73	3.55 ± 0.42	3.72 ± 0.42

AI = atherogenic index [(C12:0 + 4 × C14:0 + C16:0)/(∑MUFA + ∑(ω-6) + ∑(ω-3))]; TI = thrombogenic index [(C14:0 + C16:0 + C18:0)/((0.5 × ∑MUFA + 0.5 × ∑(ω-6) + 3 × ∑(ω-3)) + (∑(ω-3)/∑(ω-6)))].

**Table 4 molecules-25-02892-t004:** Samples collected from different geographical regions in Slovenia.

	Alpine	Dinaric	Mediterranean	Pannonian	Grand Total
2012	24	17	6	30	77
summer	12	8	3	14	37
winter	12	9	3	16	40
2013	24	17	6	30	77
summer	12	8	3	15	38
winter	12	9	3	15	39
2014	24	17	6	30	77
summer	12	8	3	16	39
winter	12	9	3	14	38
Grand Total	72	51	18	90	231

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
