# Peer review of "Fatty Acid and Stable Carbon Isotope Composition of Slovenian Milk: Year, Season, and Regional Variability"

_molecules, 2020, doi:10.3390/molecules25122892_

Round 1
Reviewer 1 Report
It is useful the calculation of Atherogenic (AI) and thrombogenic (TI) indices for the fatty acid profile.
Reviewer 2 Report
Very interesting manuscript.
The authors conducted detailed milk tests determining the fatty acid profile and described the results broken down into factors. However, the layout of the manuscript needs to be changed - material and methods (lines 389-481) should be placed on line 119, before the results and discussion.
Reviewer 3 Report
Manuscript Number: MOLECULES-831419
Title: Content and stable carbon isotope composition of fatty acids in Slovenian milk: variability related to year, season and geographical origin
Potočnik et al.
In this paper, the authors made a detailed study of fatty acid (FA) composition of 231 raw cow milk samples from Slovenia sampled in winter and summer representing four different geographical regions (Mediterranean, Alpine, Dinaric and Pannonian). The authors conclude that FA analysis content and its δ13C was effective in discriminating milk according to the year and season of production while being less successful to discriminate milk origin.
No significant plagiarism detected. Plagiarism-ChekerX (Level 3) analysis excluding title page, acknowledgements and references < 8%
To the best of my knowledge, this is an original contribution not previously published. The data presented are in agreement with the results and conclusions. This paper complements and is a logical continuation of previous work, now more specific and focused in the content of fatty acids and its compound-specific C isotope analysis in milk to study variability related to year, season and regional provenance.
In my opinion, this MS is within the scope and will certainly be of interest to MOLECULES readers. Is reasonably well written, and although the English language is fine, a careful syntax and style check will be required.
In summary, I recommend publication in MOLECULES after revision.
Specific comments
In Keywords, avoid words in the title and those with poor descriptive value i.e. origin. The authors may consider including instead some keywords related to the lipid profile or statistical analyses used Also, through the text avoid starting a sentence with an acronym.
Abstract and Introduction
The Abstract is well written, succinct and descriptive of the work done, results and conclusions. I will suggest the authors introduce here the term compound-specific isotope analysis (CSIA). This is a better-known term to describe the analytical technique used instead than GC-C-IRMS.
Line 38: Please revise the sentence and rewrite for clarity.
Line 47 & 59: Too long sentences that are difficult to follow. Please shorten and split into different sentences, one concept each.
Line 78: Write more prudently “Differences in stable C isotope abundance is mainly the result of….”
Line 82: “…plants and to enriched d13C values.”
Line 84: Write more prudently: “… condition, stable isotope composition can also encompass information about the geographical origin of dairy product useful for authenticity control.
Line 87: I will recommend using d13C, d15N, d34S, d2H or dD and d18O. Does milk water mean whey?
Line 94: “…values in milk fatty acids.”
Line 95: “Ehtesham and co-workers [22] demonstrated that δ2H values in rainwater were highly correlated with δ2H values in bulk milk and in specific fatty acids (butyric, myristic, palmitic and oleic acid).”
Line 98: The sentence “The differentiation of the region of origin was found to be 91% for milk samples.” Is vague Is this a similarity parameter? Please clarify.
Line 92 and 99: The first appearance of acronym in the main text (no abstract), please write in full.
Line 101-107: Rewrite: “Ehtesham et al. [23] studied the relationship between δ2H of fatty acids in feed and in milk components. Their results indicated that FA δ2H values in feed and that of farm water were reflected in the δ2H values of milk FAs, milk solids and milk water and that this could be used as a biogeochemical marker surrogate to environmental conditions.”
The paragraph in line 108: Please revise, order and rewrite to improve comprehension, I will suggest starting with the description of the quality seal “Selected Quality - Slovenia”, continuing exposing the need of appropriate tools to prove quality seals and then describe the specific purpose and objectives of your work.
Results and discussion
The paragraph in line 120. I think that this paragraph is not necessary. Will suggest to start directly with part 2.1. Be consistent with the tense used, will suggest past tense. The paragraphs in this section and too long and difficult to follow. Please try to separate the different findings and ideas.
The paragraph in line 218. This is written somehow carelessly and will need a complete syntax revision. Some sentences are uncompleted and the ideas not fully described or appropriately exposed.
Line 336 “For a healthier diet …”
Materials and Methods
The paragraph in line 391. The first sentence is not correct, please rewrite to reflect the real sampling. If I understand well from Table 4, a total of 231 milk samples were collected and analysed through a 3-year period (2012-2014) in winter and summer from different geographical regions in Slovenia: Mediterranean, Pannonian, Dinaric and Alpine. Please make this clear here and elsewhere including the Abstract.
Line 409: What means the reference to J.T. Baker? Is this that dichloromethane, sodium hydroxide in methanol and hexane where purchased from J. T. Baker Chemical Co.?
Tables & figures
I was unable to find the unit used for FA quantification listed in Table 1, nor that used for d13C values in Table 2 and S1 (‰ or better mUr) Please include.
In Table S2 change “Distance to the coast (m)” and Altitude I believe should be indicated in (m a.s.l.)
Enlarge figures and particularly the font size. In its present form, the figures are too small and difficult to read.
Reviewer 4 Report
Dear Authors,
in my opinion, the paper Molecules-831419 needs of some major changes in order to be take into consideration for pubblication in Molecules. I attach the pdf file with many suggested modifications.

Author Response
The answers to the reviewer’s suggestions are replied directly in the *.pdf file.

Round 2
Reviewer 4 Report
No comment